# Brownmillerites CaFeO_2.5_ and SrFeO_2.5_ as Catalyst Support for CO Oxidation

**DOI:** 10.3390/molecules26216413

**Published:** 2021-10-23

**Authors:** Pierre-Alexis Répécaud, Monica Ceretti, Mimoun Aouine, Céline Delwaulle, Emmanuel Nonnet, Werner Paulus, Helena Kaper

**Affiliations:** 1Laboratoire de Synthèse et Fonctionnalisation des Céramiques, CNRS/Saint-Gobain CREE, Saint-Gobain Research Provence, 550, Ave Alphonse Jauffret, 84306 Cavaillon, France; pierre.alexis.repecaud@gmail.com; 2ICGM, University Montpellier, CNRS, ENSCM-34095, 34296 Montpellier, France; monica.ceretti@umontpellier.fr; 3Université de Lyon, Université Claude Bernard Lyon, CNRS, IRCELYON, 2 av Albert Einstein, 69626 Villeurbanne, France; mimoun.aouine@ircelyon.univ-lyon1.fr; 4Competency Research Laboratory, Saint-Gobain Research Provence, 550, Ave Alphonse Jauffret, 84306 Cavaillon, France; Celine.Delwaulle@saint-gobain.com (C.D.); Emmanuel.Nonnet@saint-gobain.com (E.N.)

**Keywords:** oxygen mobility, CO oxidation, support interaction, phase stability, heterogeneous catalysis

## Abstract

The support material can play an important role in oxidation catalysis, notably for CO oxidation. Here, we study two materials of the Brownmillerite family, CaFeO_2.5_ and SrFeO_2.5_, as one example of a stoichiometric phase (CaFeO_2.5,_ CFO) and one existing in different modifications (SrFeO_2.75_, SrFeO_2.875_ and SrFeO_3_, SFO). The two materials are synthesized using two synthesis methods, one bottom-up approach via a complexation route and one top-down method (electric arc fusion), allowing to study the impact of the specific surface area on the oxygen mobility and catalytic performance. CO oxidation on ^18^O-exchanged materials shows that oxygen from SFO participates in the reaction as soon as the reaction starts, while for CFO, this onset takes place 185 °C after reaction onset. This indicates that the structure of the support material has an impact on the catalytic performance. We report here on significant differences in the catalytic activity linked to long-term stability of CFO and SFO, which is an important parameter not only for possible applications, but equally to better understand the mechanism of the catalytic activity itself.

## 1. Introduction

Catalytic CO oxidation on perovskites usually follows a suprafacial mechanism involving surface oxygen [1]. For reactions taking place at higher temperatures, such as methane complete oxidation, lattice oxygen from the bulk can be involved, resulting in an intrafacial mechanism [2]. While perovskites have been extensively studied for oxidation reactions, less work is available on reduced perovskites such as the Brownmillerite family ABO_2.5_ [3,4,5,6]. The first works date 40 years back, when Shin et al. studied the decomposition of NO over the two BBrownmillerites CaFeO_2.5_ and SrFeO_2.5_. At high temperatures, SrFeO_2.5_ is active, while CaFeO_2.5_ remains inactive under the studied reaction conditions. This difference has been attributed to the structural difference of the two materials, with disordered oxygen vacancies at high temperatures for SrFeO_2.5_ as opposed to the ordered vacancies in CaFeO_2.5_. More recently, Hirabayashi et al. studied calcium ferrites with a different Ca/Fe ratio for propylene combustion [7]. In their study, the best-performing material is the Brownmillerite with a Ca/Fe ratio of 1. The better performance of CaFeO_2.5_ is ascribed to the presence of active oxygen species O^2−^ on the surface. 

In this work, we want to study the influence of bulk oxygen mobility on the catalytic performance on two support material of the Brownmillerite family: CaFeO_2.5_ and SrFeO_2.5_. CaFeO_2.5_ is known as a stable line phase, and its oxygen mobility is well-described [8]. In comparison, the Brownmillerite SrFeO_2.5_ is a non-stoichiometric phase; and similar oxide phases exist, differing only in the oxygen content: SrFeO_2.75_, SrFeO_2.875_ and SrFeO_3_ are reported as distinct oxygen vacancy ordered phases. Depending on the temperature and oxygen partial pressure, one can easily shift between these modifications. This structural flexibility has been discussed to facilitate oxygen mobility already at very moderate temperatures [9]. Here, we want to compare the catalytic performance of a stable line phase (CaFeO_2.5_, CFO) with a structural flexible material (SrFeO_2.5_, SFO) for CO oxidation at low temperature. In order to discriminate between surface reactivity and reactivity as a result of structural difference, e.g., oxygen vacancy ordering, we choose two different synthesis routes, yielding different grain size and surface properties. To this end, a modified Pechini method [5] as a classic synthesis route (bottom-up approach, denoted as cp in the following), and electric arc fusion (denoted as eaf) combined with grinding (top-down approach) were utilized. The four studied samples are thus SFO_eaf and CFO_eaf for the fusion-synthesized materials and SFO_cp and CFO_cp for the materials synthesized by modified Pechini route. The latter synthesis is a high-temperature synthesis method utilized by industry, which also opens the pathway to industrialization. All samples are characterized using X-ray diffraction (XRD), X-ray fluorescence (XRF), N_2_-physisorption and ^18^O-exchange studies. The oxygen mobility under reactive gas is studied using ^18^O_2_ isotope exchange experiments, using a recently described technique (ILPOR). Finally, the catalytic performance of the Brownmillerites with 1 wt% Pt for CO oxidation is studied and the dispersion of the Pt-nanoparticles before and after catalytic testing using high-angular dark-field scanning transmission electron microscopy (HAADF-STEM).

## 2. Results

### 2.1. Physico-Chemical and Structural Properties

We first studied the physico-chemical properties of the different SFO and CFO materials. Appendix A lists the specific surface areas determined by nitrogen physisorption using the BET method. While the specific surface areas of CFO_eaf and SFO_eaf are negligibly small (below 1 m^2^/g), the surface areas are with 12 m^2^/g for CFO_cp and 13 m^2^/g for SFO_cp comparable. The two synthesis methods thus allow discriminating between the impact of the specific surface area and structure in the bulk on the oxygen mobility and catalytic performance. Additionally, the chemical compositions determined by XRF are comparable within the experimental error (Appendix A, see Appendix A). 

The X-ray diffractograms together with the refinements obtained from Rietveld refinement for CaFeO_2.5_ and SrFeO_2.5_ synthesized by a modified Pechini method and electric arc fusion are presented in Figure 1 and Table 1.

For the Brownmillerite CFO_cp and CFO_eaf, all peaks can be described to the CaFeO_2.5_ phase [10], indicating phase purity for the two samples. In the case of SFO, the situation is more complicated. Together with the main phase SrFeO_2.5_, two more phases, with different oxygen content (SrFeO_2.75_ and SrFeO_2.875_), exist. Indeed, both SFO materials consist of a mixture of the three known phases: SrFeO_2.5_ (Bragg peak positions are indicated in blue in Figure 1c,d), SrFeO_2.75_ (Bragg positions in red) and SrFeO_2.875_ (Bragg position in green). From Rietveld refinement, the contribution of each phase for SFO_eaf is 70 wt% of SrFeO_2.5_, 29 wt% of SrFeO_2.75_ and 3 wt% of SrFeO_2.875_. For SFO_cp, the phase mixture is significantly different, with only 47 wt% SrFeO_2.5_, 41 wt% SrFeO_2.75_ and 12 wt% SrFeO_2.875_. The more oxidizing conditions during air calcination of the cp synthesis leads to rather oxidized phases compared to the eaf synthesis, which enables working under more reducing conditions.

### 2.2. Catalytic Performance for CO Oxidation

We first study the bare support materials for their CO oxidation performance. Two aspects will be analyzed: the two synthesis methods allow studying the impact of the surface area on the catalytic performance. Second, the choice of materials, CFO and SFO, shows the difference in structural properties and impact of oxygen vacancy ordering and mobility on the catalytic performance. All materials were tested as-synthesized and for three consecutive runs. Figure 2 shows the light-off curve of the third run for the four samples. Both the synthesis method and the material have an influence on the catalytic CO oxidation performance. The materials synthesized by cp show better performance than the same material synthesized by eaf. The temperature at 20% CO conversion (T_20_) is 342 °C for SFO_eaf, and thus quite close to the one for CFO_cp (360 °C). For CFO_eaf, this value is with 411 °C significantly higher, and lowest for SFO_cp (318 °C). It is important to note that both SFO materials convert CO at lower temperature than any CFO material, indicating that the crystallographic properties and impact of the oxygen mobility are more important than the surface area in the studied range.

To study the reducibility of the different materials, temperature-programmed reductions under H_2_ (H_2_-TPR) were carried out. The results for CFO_cp, CFO_eaf, SFO_cp and SFO_eaf are shown in Figure 3. The reduction profiles are very similar for CFO_cp and CFO_eaf, with one large reduction peak starting between 470 °C and 800 °C and a second reduction peak starting above 800 °C. These two reduction profiles correspond to the reduction of the Brownmillerite following:CaFeO_2.5_ + ½ H_2_ → FeO + CaO + ½ H_2_O (1)
FeO + ½ H_2_ → Fe^0^ + ½ H_2_O (2)

The two materials CFO and SFO show very different behavior. In the case of SFO, we observe a very strong reduction peak between 300 °C and 575 °C. This lower reduction peak can be attributed to the reduction of Fe^4+^ present in SrFeO_2.75_ and SrFeO_2.875_ that represent roughly 50% (SFO_cp) and 30% (SFO-eaf) of the sample. Falcón et al. observed similar reduction profiles for SrFeO_2.74_ and SrFeO_2.91_ [11]. However, the intensity of this peak indicates that not only Fe^4+^ is reduced, but the reduction of Fe^3+^ starts at lower temperatures in the presence of Sr. From H_2_-TPR, we can already clearly see a difference in the behavior with respect to reducibility and therefore oxygen mobility: SFO starts to reduce at a significantly lower temperature. Additionally, we observe a strong impact of the specific surface area, notably for the SFO materials. The reduction of SFO_cp starts ca. 50 °C lower than the reduction of SFO_eaf. However, the reduction profiles are rather similar, hinting towards similar oxidation states of iron, despite the different synthesis methods. To further study the impact of the structure and surface area, the four materials were exchanged with ^18^O_2_ in the first place and in a second step, the re-exchange under air was followed using TG-MS (Figure 4).

The exchange behavior of the four materials is very similar. Below 300 °C, desorption of a species with *m*/*z* = 18 is observed. As the signal *m*/*z* = 17 is very weak, we attribute this peak rather to surface oxygen than adsorbed water. This peak is significantly more pronounced for the materials synthesized by the complexation route (Figure 4a) than for the materials synthesized by electric arc fusion (Figure 4b), further indicating that this peak can be attributed to surface exchange. For CFO_eaf and SFO_eaf, the re-exchange occurs roughly at 350 °C, as it can be seen by the decrease of the ^16^O_2_ signal and increase of the ^16^O^18^O signal. The onset is at a slightly lower temperature for SFO_eaf (357 °C for SFO_eaf and 417 °C for CFO_eaf), which accounts for a higher oxygen mobility in SFO_eaf. For the two materials synthesized by complexation, we observe an important difference by 150 °C between the onset of the re-exchange: for CFO_eaf, the onset is around 300 °C, while it is >500 °C for CFO_cp. Therefore, besides the higher specific surface area, exchange is hindered in CFO_cp, and even lower than in CFO_eaf. This might be explained by a facilitated oxygen mobility via grain boundaries in the eaf synthesized material [2]. The appearance of the *m*/*z* = 34 signal corresponding to ^16^O^18^O accounts for a monoatomic exchange according to:^16^O_2_(g) + 2 ^18^O_2_^−^(s) → ^18^O^16^O (g) + ^16^O^−^(s) (3)

The weaker signal *m*/*z* = 36 can result from a bi-atomic exchange
^16^O_2_(g) + 2 ^18^O_2_^−^(s) → ^18^O_2_ (g) + 2 ^16^O^−^(s) (4)
or via a monoatomic exchange with the reaction product from (3):^18^O^16^O (g) + 2 ^18^O_2_^-^(s) → ^18^O_2_ (g) + 2 ^16^O^−^(s) (5)

Furthermore, the release of C^16^O_2_ (*m*/*z* = 44) is observed at temperatures above 650 °C for both materials, probably due to the presence of amorphous surface carbonates. 

Therefore, the monoatomic exchange forming ^18^O^16^O is largely predominant for this type of material. Indeed, the mono-atomic exchange has already been reported for perovskite materials [12], whereas the bi-atomic exchange is predominant in ceria-based materials [13]. The results here show that besides differences in chemical compositions and structure of the two studied materials, the oxygen exchange follows the same mechanism, but at different temperatures. 

To study the influence of the chemical composition of the two compounds (CFO and SFO) on CO oxidation, we performed the reaction on the ^18^O-exchanged material, using mass spectrometry to follow the reaction products. For the experimental setup, CO is introduced in the reaction chamber as a continuous flow, while a pulsed flow is chosen for oxygen. This specific configuration, described elsewhere [5], allows at the same time following the reaction under O_2_-depleted conditions between two oxygen pulses. The experiment is conducted in the same manner as a typical light-off experiment with a continuous increase in temperature. At the moment when the reaction starts, a decrease in C^16^O is observed, and the signals for CO_2_ start to emerge. For both materials, the reaction starts around 155 °C, with C^16^O_2_ signal as the predominant signal, indicating that the reaction takes also place without ^18^O from the support (Figure 5). However, in the case of SFO_eaf, we observe at the same time the presence of C^16^O^18^O, showing that both, the support and gas-phase oxygen contribute to the reaction. For CFO_eaf, this contribution is observed at a higher temperature (185 °C) than the reaction onset, indicating hindered reaction kinetics in the lower temperature range. 

We also note that the reaction starts at a lower temperature than the oxygen mobility observed by H_2_-TPR and TG-MS (*vide sopra*) on the exchanged materials. In both cases, the materials are reactive above 300 °C. The lower reactivity here corresponds thus to surface reactivity. Moreover, since we work with O_2_ pulses, between two pulses we have reducing conditions, due to the presence of CO, which is a strong reductor and which interferes with the ^18^O oxygen exchange reaction. 

When looking at the oscillatory behavior of the four materials (Appendix A), the signal of ^16^O_2_ is in phase with the CO_2_ formed during the reaction for SFO_cp, SFO_eaf and CFO_eaf. Interestingly, we observe a different behavior for CFO_cp. Here, at the beginning of the reaction, the ^16^O_2_ signal is not in phase with the CO_2_ signal, and only after ca. 20 min homogeneity between the two phases is reached. Signals that are in phase indicate fast reaction kinetics, e.g., no observable delay between consumption of reactants (CO) and release of products (CO_2_). When these signals are out of phase, the kinetics on one or more of the different steps involved (adsorption/migration/dissociation/reaction/desorption) are slower. The slower reaction kinetic observed for CFO_cp indicates lower surface reactivity compared to the SFO materials and even CFO_eaf. This further confirms that CFO an SFO are inherently different materials, with SFO showing higher oxygen mobility contributing as soon as the reaction starts to the reaction.

In order to identify the stability of the different phases, XRD analysis was carried out after the ILPOR experiment (Figure 6). Indeed, the here-studied materials show very different phase stabilities under the applied reaction conditions. In particular, the materials synthesized using the cp route turned out to not be very stable, and decompose into several phases (formation of carbonates and iron oxide, next to the Brownmillerite). The materials synthesized by electric arc fusion are more stable, not showing any trace of decomposition, probably due their better intrinsic crystallinity.

In order to decrease the reaction temperature and study the stability of the material as a functionalized catalyst, the materials were impregnated with 1 wt% Pt. Three consecutive runs were carried out to study the stability of the materials upon cycling under reaction conditions. Figure 7 shows the first and third run for Pt/SFO_cp (Figure 7a) and Pt/SFO_eaf (Figure 7b). We notice an important difference between Pt/SFO_cp and Pt/SFO_eaf upon cycling. While Pt/SFO_cp shows better performance with a CO conversion at significantly lower temperature during the first run (T_20_ = 182 °C compared to 230 °C for Pt/SFO), the T_20_ increases to 246 °C for Pt/SFO_cp during the 3rd run. Pt/SFO_eaf remains rather stable during the three runs (T_20_ = 238 °C). These values are well above the CO oxidation performance of other well-known Pt catalysts tested, such as Pt/CeO_2_ [14], but similar to perovskites of similar composition, e.g., Pt/SrTi_0.65_Fe_0.35_O_3−δ_ [15]. The deactivation observed in particular for Pt/SFO_cp could be due to platinum nanoparticle growth, or changes in the support, as already noted after the ILPOR experiment (*vide sopra*).

The possible growth of platinum nanoparticles can be studied using HAADF-STEM analysis before and after catalytic testing (Figure 8).

Despite the low specific surface area of SFO_eaf, platinum is well-dispersed on the surface with a particle size of 1.3 +/−0.3 nm (Figure 8b). This particle size is also very similar to the one of Pt/SFO_cp before testing, with an average particle size of 1.1 +/− 0.4 nm (Figure 8a). The difference observed in the CO oxidation performance is thus due to the impact of the support, and not the Pt particle size. The spent catalysts show rather different behavior. For Pt/SFO_eaf (Figure 8d), the particle size is nearly unchanged when compared to the fresh samples (average particle size of 1.4 +/− 0.3 nm). In comparison, the particle size increases for Pt/SFO_cp from 1.1 +/− 0.4 nm to 1.7 +/− 0.5 nm (Figure 8c). The deactivation of Pt/SFO_cp upon cycling can thus be partially explained by the growth in Pt particle size. Again, we observe higher stability for the eaf-synthesized samples with respect to stability of the Pt-nanoparticles, but also the support materials. 

## 3. Materials and Methods

CaFeO_2.5_ and SrFeO_2.5_ were prepared by a recently described method [5], herein called complexation method. Stoichiometric amounts of Ca(NO_3_)_2_·4H_2_O (Sigma-Aldrich, Steinheim, Germany >99%) or Sr(NO_3_)_2_ (Alfa-Aesar, Kandel, Germany, >99%) and Fe(NO_3_)_3_·9H_2_O (Alfa Aesar, Kandel, Germany, >98%) were dissolved in water. Then, citric acid (Sigma-Aldrich, >99.5%) was added and the solution was stirred for 2 h. The molar ratio of citric acid to the total amount of cations was kept to one. The pH value was adjusted to pH = 3 using 37% ammonia solution (Merck, Darmstadt, Germany, 28%), followed by the addition of ethylene glycol (Sigma-Aldrich, Saint-Louis, MO, USA, >99.8%). The solution was evaporated at 90 °C on a hot plate until jellification. The gel was then quickly fired at 400 °C in air to obtain a powder. The as-obtained powder was calcined by heating under atmosphere (synthetic air for CaFeO_2.5_ and pure argon for SrFeO_2.5_) at 2 °C/min up to 600 °C and kept for 6 h at this temperature. The powders were finally crushed in a mortar.

To synthesize CaFeO_2.5_ and SrFeO_2.5_ by EAF, an Electric Arc Furnace with graphite electrodes was used. Stoichiometric amounts of CaCO_3_ (Omya SAS, Orgon, France) or SrCO_3_ were mixed with Fe_2_O_3_ (Colorey E172, Lozanne, France). By adjusting the electric potential and electric current, the powders were melted at 1600 °C. The liquid mix was then quenched in air giving agglomerates with a grain size of several centimeters.

These agglomerates were first crushed for 72 h in a low-energy ball-mill. To further reduce the grain size, the powders were ground in isopropanol (Sigma-Aldrich, >98%) for 1 h at high energy (1000 rpm) using an attrition mill. 

The obtained powders were impregnated with 1 wt.% of platinum using classical wet impregnation. To a suspension of the bare powders in isopropanol (Sigma-Aldrich, >98%), the proper amount of Pt(NO_3_)_2_ (Heraeus, Hanau, Germany >99%) was added under stirring. After 2 h the suspension was placed in a rotavapor (Heidolph) 75 rpm, 130 mbar, 50 °C until obtaining a dry powder. The resulting powder was then calcined at 500 °C during 2 h in air (heating ramp: 10 °C.min^−1^).

^16^O/^18^O isotope exchange was performed in a quartz reactor. In a typical enrichment, 300 mg of powder treated under vacuum and then exposed to 1 bar of ^18^O_2_ (Eurisotop, Saint Aubin, France) at 400 °C during 30 min. The vacuum ^18^O_2_ atmosphere steps were repeated three times leading to an enrichment of ca. 65%.

Elementary analyses with X-ray fluorescence (Panalytical) were performed to verify the chemical composition of each composition. The powders were mixed with LiB4 prior to melting and analysing.

Specific surface areas were measured by N_2_-physisorption using the Brunauer-Emmett-Teller method. Each sample was degassed under vacuum for 1 h at 180 °C. The isotherms were measured at 78 K on a 2820 Tristar (Micromeritics, GA, USA).

In order to quantify the platinum impregnation in term of platinum dispersion, high-angle annular dark-field scanning transmission electron microscopy (HAADF-STEM) pictures were taken with a FEI TITAN ETEM (Thermo Fisher). The particle size distribution was determined by measuring the size of at least 300 particles, assuming spherical particles according to [16].
(6)$$D\left(\%\right)=\frac{6*M_{pt}*d_{va}*100}{a*p*N_{a}}$$
where ***M_Pt_*** is the atomic mass of platinum, ***d_va_***—the volume–area mean diameter, ***a***—the surface area occupied by one platinum atom, ***ρ***—the platinum mass density and ***N_a_***—the Avogadro number. ***d_va_*** is described by Equation (7) as
(7)$$d_{va}=\frac{\sum^{\;}n_{i}*d_{i}^{3}}{\sum^{\;}n_{i}*d_{i}^{2}}$$
where *n_i_* is the number of particles with a diameter *d_i_*.

The reducibility of the materials was studied using H_2_-TPR experiments using an automated AutoChem II 2920 from Micromeritics. A 200 mg sample was introduced on quartz wool in a U-shaped quartz reactor. Prior to H_2_-TPR, an oxidative pre-treatment was carried out by heating the sample to 800 °C under oxygen (oxygen flow: 30 mL/min, heating ramp: 10 °C/min) The samples were cooled down under helium to room temperature. In the second reductive step, the sample was heated under 3%H_2_/97% argon (30 mL/min) at 10 °C/min up to 1000 °C and kept for one hour. The H_2_ uptake was measured with a thermal conductivity detector (TCD).

Laboratory X-ray powder diffraction patterns were collected on a Panalytical Analytical X’Pert PRO diffractometer equipped with a X’Celerator detector and operating in Bragg–Brentano geometry (/2) using CuK_1,2_. Rietveld structure refinement was carried out through the Fullprof software [17]. (The complete FULLPROF suite can be obtained from http://www.ill.eu/sites/fullprof/index.html, accessed on 31 August 2021).

CO oxidation was carried out in a fixed bed through-flow reactor. The U-shaped quartz reactor was filled with 200 mg of sample. Prior to the catalytic test, each sample was pre-treated at 200 °C during 2 h under 40% H2/60% helium (10 L.h^−1^). For CO oxidation, a mixture of 6000 ppm CO and 10,000 ppm of O_2_ (diluted in He) with a GHSV of 200 000 h^−1^ was used. The sample was heated from room temperature to 500 °C with a heating rate of 2 °C/min. The gas products were detected and quantified with a micro-gas chromatography (SRA) equipped with a TCD detector. 

Isotope Labelling Pulse temperature programmed Oxidation Reaction (ILPOR) experiments were carried out using an automated catalyst characterization system (Autochem 2920, Micromeritics) coupled with a quadrupole mass spectrometer (QMS, ThermoStar TM GSD 301T, PFEIFFER VACUUM). A 200 mg ^18^O-enriched sample was loaded in a U-shaped reactor and stabilized at 40 °C in ^16^O_2_ atmosphere (30 mL/min) for 1 h. The sample was then heated from room temperature to 500 °C at 2 °C/min. During the heating program, the sample was under a constant flux of CO (30 mL/min, 10,000 ppm) to which pulses of ^16^O_2_ were injected through a loop valve with an inside volume of 0.6407 mL. The injection process was repeated every 60 s. The evolution of the different oxygen-containing species (*m*/*z* = 16 for ^16^O, *m*/*z* = 17 for ^16^OH, *m*/*z* = 18 for ^18^O or H_2_^16^O, *m*/*z* = 19 for ^18^OH, *m*/*z* = 20 for H_2_^18^O, *m*/*z* = 28 for C^16^O, *m*/*z* = 30 for C^18^O, *m*/*z* = 32 for ^16^O_2_, *m*/*z* = 34 for ^18^O_16_O, *m*/*z* = 36 for ^18^O_2_, *m*/*z* = 44 for C^16^O_2_, *m*/*z* = 46 for C^16^O^18^O and *m*/*z* = 48 for C^18^O_2_) were recorded on the QMS during the temperature-programmed oxidation of CO.

## 4. Conclusions

Our results clearly demonstrate the importance of the synthesis conditions for a given system, together with the existence or not of oxygen non-stoichiometry to better understand its catalytic activity and stability in oxidation reactions. Furthermore, the here-shown results nicely demonstrate the importance of analyzing a catalytic system as a whole, including support and active noble metal phase. We differentiate between both the impact of the surface area and the crystallographic structure by employing two synthesis methods (one top-down and one bottom-up approach) and CFO and SFO with inherent different crystallographic structure. Indeed, not only surface properties, but more importantly, structural properties such as facilitated oxygen ion mobility in SFO materials contribute significantly to the CO oxidation performance of these materials. H_2_-TPR analysis and TG-MS on ^18^O-exchanged CTF and SFO demonstrate that SFO shows superior reducibility and oxygen mobility in the bulk. The facilitated oxygen mobility directly translates into superior CO oxidation performance, indicated by CO oxidation taking place at lower temperatures for both SFO materials compared to the corresponding CFO material. The T_20_ of SFO_eaf is 69 °C lower than CFO_eaf, and the same trend, albeit with a slightly lower impact, is observed for the two materials synthesized by the complexation route. Here, the difference between the T_20_ is 42 °C between SFO_cp and CFO_cp. An important difference in terms of bulk oxygen participation comparing CFO and SFO is observed during the ILPOR technique, allowing to evidence a temperature difference for bulk oxygen to participate in the catalytic reaction of almost 200 °C above the surface reaction to set in. We also note an important difference in stability for the two synthesis methods. Pt/SFO_eaf is more stable with respect to the CO oxidation performance upon cycling than Pt/SFO_cp, which degrades with respect to the Pt particle size distribution, and phase purity of the support phase. Therefore, materials synthesized using top-down approaches such as the eaf synthesis show interesting features as catalyst support, in particular with respect to long term stability, which is an important aspect for application.

## Figures and Tables

**Figure 1 molecules-26-06413-f001:**
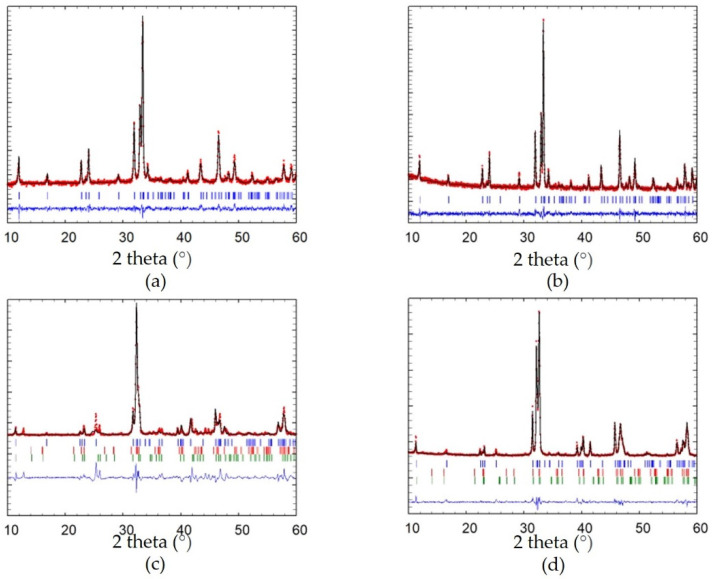
XRD pattern and Rietveld refinement of (**a**) CFO_cp, (**b**) CFO_eaf and (**c**) SFO_cp, (**d**) SFO_eaf. Red curves: measured data, black curve: Rietveld refinement, blue line: difference. Vertical bars indicate the Bragg peak position: the blue ones correspond to the main phase CaFeO_2.5_ (in a,b), SrFeO_2.5_ (in c,d), while the red and green ones in (c) and (d) correspond to SrFeO_2.75_ and SrFeO_2.87_, respectively).

**Figure 2 molecules-26-06413-f002:**
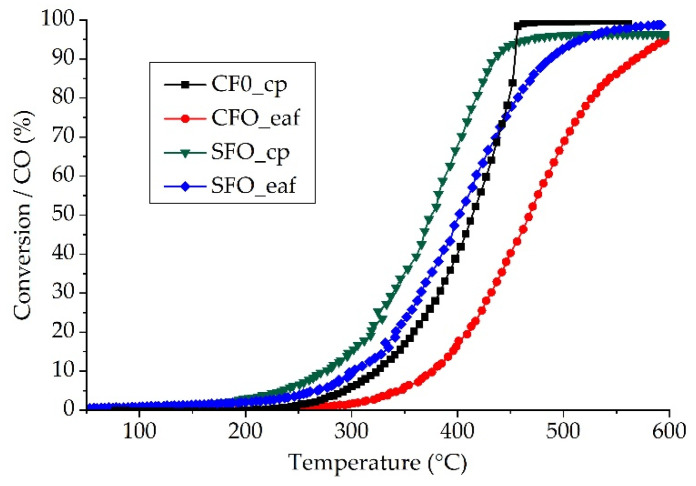
CO oxidation of CFO and SFO synthesized by complexation and fusion synthesis. For clarity, the stabilized third runs are shown. Reaction conditions: 6000 ppm CO, 10,000 ppm O_2_, 200 mg catalyst, total flow: 10 L/h.

**Figure 3 molecules-26-06413-f003:**
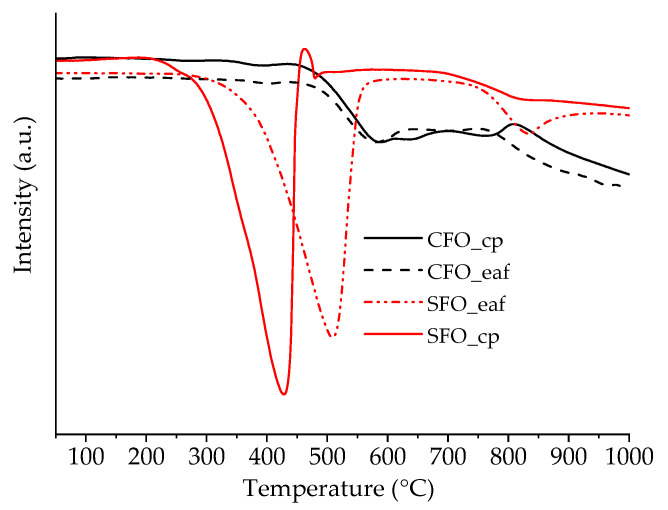
H_2_-temperature-programmed reduction of CFO_cp, CFO_eaf, SFO_cp and SFO_eaf.

**Figure 4 molecules-26-06413-f004:**
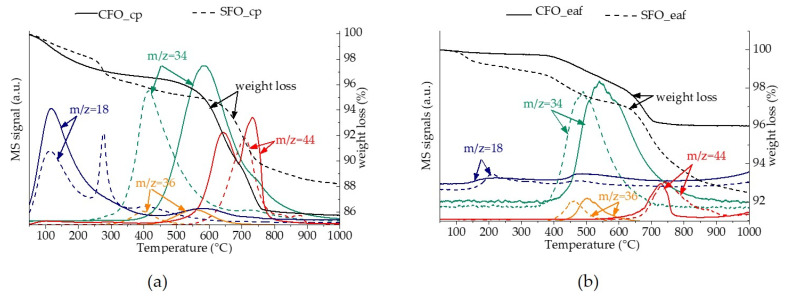
TG-MS analysis under air on ^18^O-exchanged CFO (solid lines) and SFO (dashed lines). (**a**) CFO and SFO synthesized by the complexation route. (**b**) CFO and SFO synthesized by electric arc fusion. Black curves: TG analysis, colored curves: mass spectrometry analysis.

**Figure 5 molecules-26-06413-f005:**
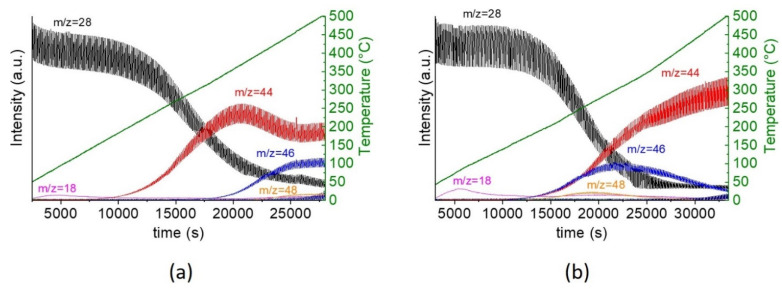
Pulsed CO oxidation on ^18^O-exchanged CFO_eaf (**a**) and SFO_eaf (**b**). The mass signals correspond to the following gases: *m*/*z* = 18: water or surface oxygen, *m*/*z* = 28: CO, *m*/*z* = 44: C^16^O^16^O, *m*/*z* = 46: C^18^O^16^O and *m*/*z* = 48: C^18^O^18^O. For clarity, the O_2_ signal *m*/*z* = 32 is not shown. The experiment is carried out under oxidative conditions, so the *m*/*z*=32 signal is stable.

**Figure 6 molecules-26-06413-f006:**
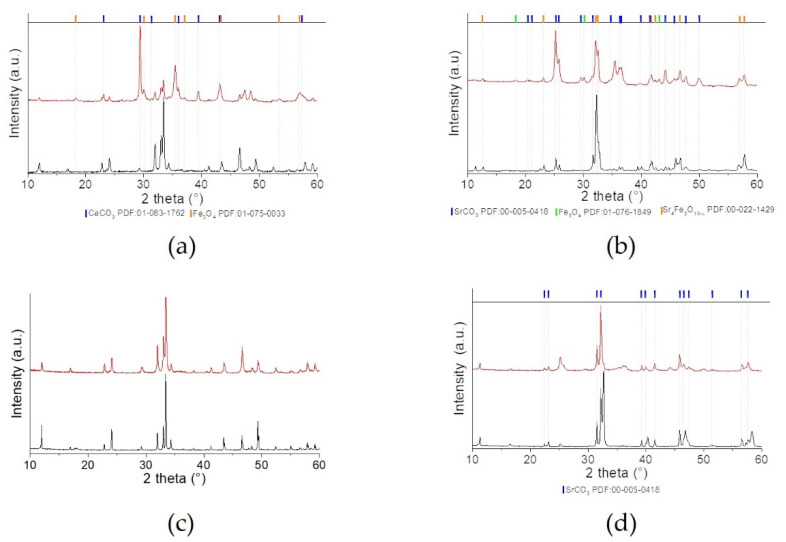
XRD patterns of (**a**) CFO_cp, (**b**) SFO_cp, (**c**) CFO_eaf and SFO_eaf (**d**) as-synthesized (black pattern) and after ILPOR test (red pattern).

**Figure 7 molecules-26-06413-f007:**
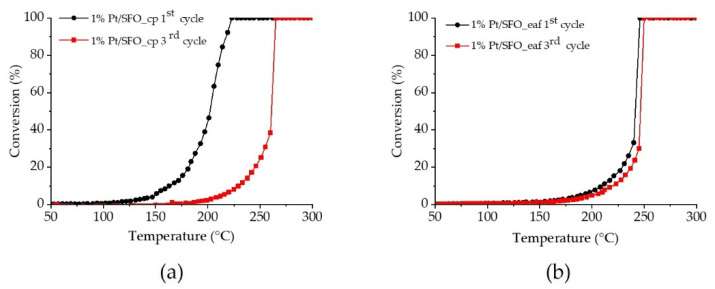
CO oxidation on Pt/SFO synthesized by complexation (**a**) and fusion synthesis (**b**). For clarity, the first and third runs are shown. Reaction conditions: 6000 ppm CO, 10,000 ppm O_2_, 200 mg catalyst, total flow: 10 L/h.

**Figure 8 molecules-26-06413-f008:**
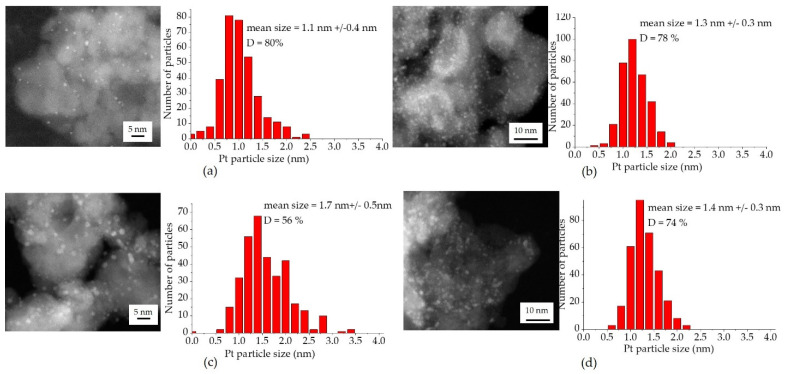
HAADF-STEM analysis of platinum-supported catalysts before and after CO oxidation. (**a**,**c**) Pt/SFO_cp before and after CO oxidation; (**b**,**d**) Pt/SFO_eaf before and after CO oxidation.

**Table 1 molecules-26-06413-t001:** Refinement data and phase composition obtained from Rietveld refinement for CFO_cp, CFO_eaf, SFO_cp and SFO_eaf. Lattice parameters a, b and c are given in Å.

Sample	Phase	Wt%	Space Group	a	b	c
CFO_cp	CaFeO_2.5_	100	*Pnma*	5.433 (2)	14.776 (7)	5.604 (2)
CFO_eaf	CaFeO_2.5_	100	*Pnma*	5.430 (4)	14.77 (1)	5.597 (4)
SFO_cp	SrFeO_2.5_	47	*Icmm*	5.638 (6)	15.55 (2)	5.517 (5)
	SrFeO_2.75_	41	*Cmmm*	11.10 (4)	7.66 (1)	5.52 (2)
	SrFeO_2.875_	12	*I4/mmm*	10.97 (2)	10.97 (2)	7.63 (2)
SFO_eaf	SrFeO_2.5_	70	*Icmm*	5.638 (6)	15.55 (2)	5.517 (5)
	SrFeO_2.75_	27	*Cmmm*	11.10 (4)	7.66 (1)	5.52 (2)
	SrFeO_2.875_	3	*I4/mmm*	10.97 (2)	10.97 (2)	7.63 (2)

## Data Availability

The data presented in the study are available upon request from the corresponding authors.

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
