# Peer review of "Brownmillerites CaFeO2.5 and SrFeO2.5 as Catalyst Support for CO Oxidation"

_molecules, 2021, doi:10.3390/molecules26216413_

Round 1

Reviewer 1 Report

The paper of Repecaud et al. deals with the synthesis of SrFeO2.5 and CaFeO2.5 prepared by electric arc fusion and Pechini method and tested for the catalytic CO oxidation. The manuscript display a rich characterization of the materials involving multiple methods thus allowing accurate study of the structure-property relationships.  Due to the scarce utilization of brownmillerite phases as oxidation catalysts, this paper represent interesting results and is therefore highly suitable for publication in this journal after minor revision.

Page 2 Line 71: The sample labels are reported without definition. This however are given few line below.  Please report the label definition at the right place.

CO oxidation experiments: The activation energy from the light-off curves should be determined to better evaluate the activity of the different materials.

Can the authors comment the surface chemistry of the materials investigated? How differ the surface states of iron of the materials prepared by Pechini and electric arc fusion? Do the Pechini samples have more carbonate species than those obtained by electric arc fusion?

Author Response

The paper of Repecaud et al. deals with the synthesis of SrFeO2.5 and CaFeO2.5 prepared by electric arc fusion and Pechini method and tested for the catalytic CO oxidation. The manuscript display a rich characterization of the materials involving multiple methods thus allowing accurate study of the structure-property relationships.  Due to the scarce utilization of brownmillerite phases as oxidation catalysts, this paper represent interesting results and is therefore highly suitable for publication in this journal after minor revision.

Page 2 Line 71: The sample labels are reported without definition. This however are given few line below.  Please report the label definition at the right place.

We used the sample labels for the first time during the introduction.

CO oxidation experiments: The activation energy from the light-off curves should be determined to better evaluate the activity of the different materials.

Can the authors comment the surface chemistry of the materials investigated? How differ the surface states of iron of the materials prepared by Pechini and electric arc fusion? Do the Pechini samples have more carbonate species than those obtained by electric arc fusion?

The main difference between the two samples is indeed the specific surface area, which is notably higher for the materials synthesized by the Pechini method. This clearly leads to higher amount of surface carbonates, as also seen by more pronounced CO2 signal in Figure 4 (m/z= 44 corresponds to CO2). Even though mass spectrometry is a qualitative technique in our case, the relative intensities can be exploited.

Reviewer 2 Report

This manuscript by Repecaud et al presented an interesting study by comparison of CaFeO2.5 and SrFeO2.5 as independent catalysts materials or catalysts for CO oxidation reactions demonstrating a significant discovery of the importance of preparation methods and formulations. The results seems novel and sound however there are some concerns I would recommend the authors to address detailed as below:

1) The authors presented a detailed weight percentage of three SrFeOx species in table 1, I am curious how it is determined. Please clarify and explain.

2. in Figure 4, both CFO_cp and SFO-cp shows a baseline at zero which makes more sense that when there is no such species evolved. I am wondering why the baseline in the case of CFO_ear and SFO_ear are above zero for m/z=18 and 34. Please explain.

3. in Figure 7. Pt/SFO_cp 1st cycle shows the best catalytic performance but decay significantly after 3 cycles. I would like the authors to explain the standard errors in each points and whether the decay occurs during CO testing or simply after CO testing and rerun. 

4. In Figure 8, the size is lacking of standard deviation and corresponding TEM image. 

5. minor thing: in line 191, the sentence is not complete. 

in line 241, "growht" should be "growth"

Author Response

This manuscript by Repecaud et al presented an interesting study by comparison of CaFeO2.5 and SrFeO2.5 as independent catalysts materials or catalysts for CO oxidation reactions demonstrating a significant discovery of the importance of preparation methods and formulations. The results seems novel and sound however there are some concerns I would recommend the authors to address detailed as below:

  • The authors presented a detailed weight percentage of three SrFeOx species in table 1, I am curious how it is determined. Please clarify and explain.Answer : From Rietfeld analysis

  1. in Figure 4, both CFO_cp and SFO-cp shows a baseline at zero which makes more sense that when there is no such species evolved. I am wondering why the baseline in the case of CFO_ear and SFO_ear are above zero for m/z=18 and 34. Please explain.

Answer : I am not sure I understand the question. The signals from the mass spectrometer are not calibrated. As such, the signals of the different m/z do not necessarily start at a baseline at zero. The values are relative values, and only the difference with respect to the start signal are regarded.

  1. in Figure 7. Pt/SFO_cp 1st cycle shows the best catalytic performance but decay significantly after 3 cycles. I would like the authors to explain the standard errors in each points and whether the decay occurs during CO testing or simply after CO testing and rerun. 

Answer: The catalyst deactivates during the run, probably due to changes in the Pt-particle size, as explained in the article. The catalysts were run immediately one after another, without any pretreatment or exposure to different atmosphere (air) between two runs.

  1. In Figure 8, the size is lacking of standard deviation and corresponding TEM image. 

This has been corrected

5. minor thing: in line 191, the sentence is not complete. 

We realized that the sentence was cut by a paragraph, but continues after Figure

in line 241, "growht" should be "growth"

This has been corrected